# Hippocampal pattern completion is linked to gamma power increases and alpha power decreases during recollection

Bernhard P Staresina[1]*, Sebastian Michelmann[1†], Mathilde Bonnefond[2†], Ole Jensen[2], Nikolai Axmacher[3,4,5], Juergen Fell[6]

[1]School of Psychology, University of Birmingham, Birmingham, United Kingdom; [2]Donders Institute for Brain, Cognition and Behaviour, Radboud University Nijmegen, Nijmegen, Netherlands; [3]Department of Neuropsychology, Ruhr University Bochum, Bochum, Germany; [4]Institute of Cognitive Neuroscience, Faculty of Psychology, Ruhr University Bochum, Bochum, Germany; [5]Faculty of Psychology, Ruhr University Bochum, Bochum, Germany; [6]Department of Epileptology, University of Bonn, Bonn, Germany

**Abstract** How do we retrieve vivid memories upon encountering a simple cue? Computational models suggest that this feat is accomplished by pattern completion processes involving the hippocampus. However, empirical evidence for hippocampal pattern completion and its underlying mechanisms has remained elusive. Here, we recorded direct intracranial EEG as human participants performed an associative memory task. For each study (encoding) and test (retrieval) event, we derived time-frequency resolved representational patterns in the hippocampus and compared the extent of pattern reinstatement for different mnemonic outcomes. Results show that successful associative recognition (AR) yields enhanced event-specific reinstatement of encoding patterns compared to non-associative item recognition (IR). Moreover, we found that gamma power (50–90 Hz) increases – in conjunction with alpha power (8–12 Hz) decreases not only distinguish AR from IR, but also correlate with the level of hippocampal reinstatement. These results link single-shot hippocampal pattern completion to episodic recollection and reveal how oscillatory dynamics in the gamma and alpha bands orchestrate these mnemonic processes.

**\*For correspondence:**
b.staresina@bham.ac.uk

[†]These authors contributed equally to this work

**Competing interests:** The authors declare that no competing interests exist.

## Introduction

A subtle reminder can bring back a wealth of rich and detailed memories. This ability to mentally travel back in time upon encountering an external or internal cue ('episodic memory') is arguably one of the main pillars of cognition and behaviour. How does the brain accomplish this feat? Ignited by neuropsychological work and corroborated by animal models and human neuroimaging, converging evidence points to the medial temporal lobe (MTL), and the hippocampus in particular, as the key brain region supporting episodic memory (*Davachi, 2006*; *Scoville and Milner, 1957*; *Squire, 1992*). However, the mechanistic processes through which the hippocampus enables the vivid recollection of past experiences are less well understood.

Based on the physiological properties of the hippocampal CA3 subregion, computational models have proposed 'pattern completion' as the central mechanism underlying successful recollection. Specifically, dense recurrent connections among CA3 pyramidal cells are thought to allow, after a single exposure, for auto-associative reinstatement of a previous learning pattern upon receiving a retrieval cue (*Marr, 1971*; *Rolls, 2016*). Consistent with a role for pattern completion, selective knock-out of the CA3 NMDA receptor was shown to result in impaired memory on a Morris water

maze when only partial environmental cues were available, i.e. when performance presumably relied more strongly on successful pattern completion (*Nakazawa et al., 2002*) (see also [*Neunuebel and Knierim, 2014*]). Moreover, recent optical imaging and optogenetic studies in mice were able to show that activation in local hippocampal cell assemblies recorded during contextual fear conditioning is reinstated during later re-exposure to the learning context (*Tayler et al., 2013*) and that experimental activation of those assemblies leads to the expression of the learned behaviour (*Liu et al., 2012*). However, whether local hippocampal pattern completion indeed emerges after single-shot (episodic) learning and whether it underlies the mnemonic expression of recollection in humans remains an open question.

The recent advent of multivariate analytical tools in neuroimaging has provided a potentially sensitive method for capturing pattern completion processes in human memory paradigms (*Norman et al., 2006*). In brief, multivariate representational patterns (e.g., voxel intensities across an anatomically defined region in a functional magnetic resonance (fMRI) study) can be derived for a particular encoding event, akin to a 'neural fingerprint' of a given learning experience. During a subsequent memory test, the extent to which this pattern is reinstated as a function of memory performance can then be assessed. Indeed, using this analytical approach, a series of recent fMRI studies have furnished evidence for pattern reinstatement in category-specific neocortical brain regions during successful recollection (*Bosch et al., 2014*; *Horner et al., 2015*; *Ritchey et al., 2012*; *Staresina et al., 2012b*; *Tompary et al., 2016*). But although overall BOLD changes in the hippocampus were found to co-vary with cortical activation/reinstatement, none of these fMRI studies has found hippocampal reinstatement to (i) result from single-shot learning, (ii) selectively support recollection/associative memory and (iii) be event-specific. More importantly, even when applying more lenient criteria for pattern completion, fMRI studies lack the temporal resolution to identify the temporal/oscillatory mechanisms underlying this process, leaving a significant gap in our understanding of how hippocampal contributions to episodic memory are orchestrated. For instance, high-frequency gamma power increases in the hippocampus have consistently been related to successful memory encoding and retrieval (*Burke et al., 2014*; *Hanslmayr et al., 2016*), which raises the possibility that gamma power (and/or power changes in other frequencies) might be directly linked to hippocampal pattern completion.

In this study, we used the rare opportunity to record electrophysiological activity directly from the hippocampus of pre-surgical epilepsy patients. In an associative memory paradigm (*Figure 1*), participants encoded trial-unique concrete nouns paired with one of four different associative details: the colour blue, the colour red, an indoor scene or an outdoor scene. During retrieval, previously seen (old) and previously unseen (new) nouns were presented. Participants indicated, with a single button press, whether they thought the noun was old or new and in case they thought it was old, whether they also remembered the target association (used to operationalize recollection). To quantify pattern completion, we then correlated the dynamic time-frequency patterns during retrieval events with the corresponding patterns during encoding events (*Manning et al., 2011*; *Yaffe et al., 2014*; *Zhang et al., 2015*), both for successful associative recognition (AR) and non-associative item recognition (IR).

## Results

### Behavioural results

Response distributions for memory retrieval are listed in *Table 1*. For all subsequent analyses, trials in which participants indicated they do not remember the associated detail ('?' responses) and trials in which an incorrect target response was given were combined to an Item Recognition (IR) condition and contrasted to trials in which the correct target response was given (Associative Recognition, AR). Thus, the word was correctly recognized in both conditions, with the critical difference that participants additionally remembered the correct associative detail in the AR condition. After artefact rejection, the AR condition contained an average of 87 trials (range 43–141) and the IR condition contained an average of 81 trials (range 32–137) (t(10) = 0.30, P = 0.767).

Response latencies were significantly shorter for AR compared to IR (1.91 s (±0.13 s) vs. 2.14 s (±0.12 s), t(10) = 2.88, P = 0.016). When considering '?' responses and incorrect target responses

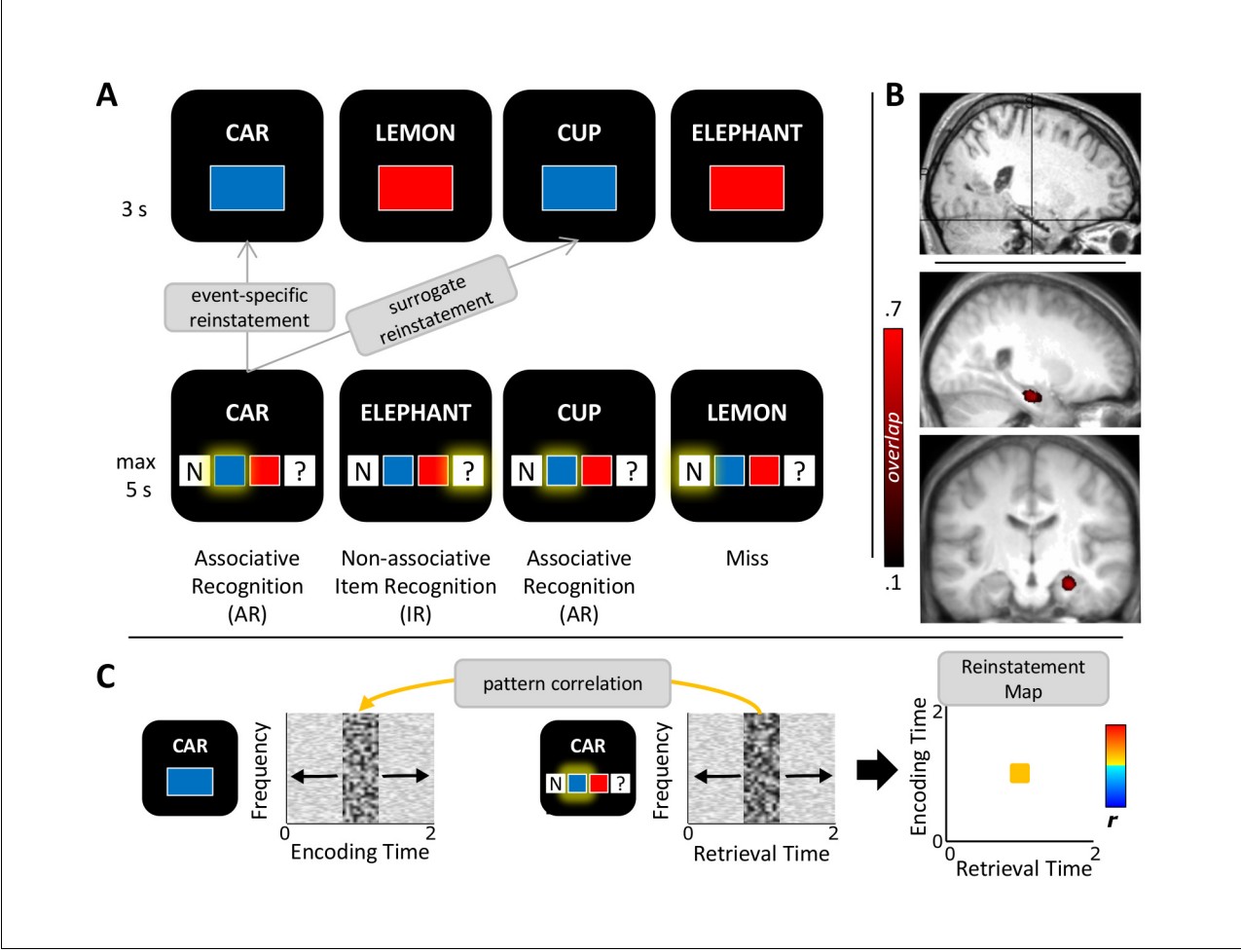

**Figure 1.** Study protocol. (**A**) Experimental paradigm. During encoding (*top*), participants saw nouns together with one of two colours or one of two scenes (not shown) and indicated whether the combination was plausible or implausible. During retrieval (*bottom*), the same nouns along with previously unseen nouns were shown and participants indicated their memory for both the noun and the association. Pattern completion was operationalized as the event-specific reinstatement during successful associative recognition (AR) compared to non-associative item recognition (IR) and compared to surrogate data representing the similarity with all other trials in which the same association was recollected. (**B**) Anterior hippocampus electrode selection. *Top*: MNI-normalized post-implantation MRI of a single participant, with cross-hair placed on the selected contact. *Bottom*: Saggital and coronal mean pre-implantation MRI across 11 participants (MNI-normalized). Heat map shows the proportional overlap of 5 mm-radius spheres centred on each participant's selected electrode. (**C**) Schematic overview of the reinstatement analysis. For each trial, retrieval and encoding patterns were correlated via a sliding 400 ms window encompassing relative power changes from 2–100 Hz (1-Hz steps from 2–29 Hz, 5-Hz steps from 30–100 Hz). Each instance of correlating a frequency x time encoding pattern with a frequency x time retrieval pattern results in a single correlation bin in a trial-specific reinstatement map (*right*). These maps were then averaged for each memory condition and taken forward to random-effects analyses across participants.

separately, their response latencies did not differ reliably (t(10) = 1.12, P = 0.290), whereas correct AR responses were significantly faster than both (both t(10) > 2.29, P<0.05).

To increase statistical power, we collapsed data across colour and scene blocks in this study. As shown in *Table 2*, during encoding, 'plausible' and 'implausible' responses were well balanced across participants for both colour and scene blocks, with no main effect of block type, no main effect of response and no interaction between the two factors (all F(1,10) < 0.60, P>0.461). For retrieval, proportions of memory conditions were again balanced across colour and scene blocks, with an average of 51% (±6%) AR trials vs. 49% (±6%) IR trials for colour blocks and an average of 52% (±5%) AR trials vs. 48% (±5%) IR trials for scene blocks, without a main effect of condition nor a condition x block type interaction (both F(1,10) < 0.15, P>0.706). Finally, when including the factor

**Table 1.** Behavioural results. **a.** Average (and SEM) proportion of hits and correct rejections out of all old and new test nouns, respectively. **b.** Average (and SEM) proportion of associative memory performance out of all hits.

| a. recognition memory | |
| --- | --- |
| hit | 0.76 (0.04) |
| correct rejection | 0.79 (0.05) |
| **b. associative memory** | |
| association correct | 0.52 (0.05) |
| "don't know" association | 0.32 (0.07) |
| association incorrect | 0.16 (0.03) |

Category (colours, scenes) and Memory (AR, IR) in a repeated measures ANOVA on response latencies, there was only a main effect of Memory ($F_{(1,10)} = 6.75$, $P = 0.027$), without a Category main effect ($F_{(1,10)} = 1.54$, $P = 0.243$) or a Category x Memory interaction ($F_{(1,10)} = 0.68$, $P = 0.428$).

## Hippocampal pattern completion

For each of the 11 participants, we selected a contact in the anterior hippocampus (*Figure 1B*; Materials and methods). Pattern completion during episodic memory retrieval was assessed as follows (*Figure 1C*): First, a representational pattern was defined as a 400 ms time window (10 ms temporal resolution) including the proportional power changes from 2 to 100 Hz (1-Hz steps from 2–29 Hz, 5-Hz steps from 30–100 Hz) relative to a 500 ms prestimulus baseline window (in keeping with our standard time-frequency analyses, see below). A representational pattern centred on a given time point thus consisted of 43 x 41 frequency values. Note however that the same results were observed when using the 43 frequencies only (i.e. not extended across time), when extending the time window from 400 ms to 500 ms, or when averaging across time points (*Yaffe et al., 2014*); *Figure 4—figure supplement 1*). Next, using a sliding window (10 ms steps), the Spearman correlations between a given trial's encoding patterns across time and its retrieval patterns across time were calculated, resulting in an encoding time x retrieval time reinstatement map for each trial. For each participant, these trial-specific reinstatement maps were averaged across all AR and IR trials, respectively, and taken forward to second-level random-effects analyses.

One possible caveat if observing greater reinstatement for AR than for IR may be that participants might direct their attention more strongly to the target association displayed on the screen during retrieval, yielding a stronger perceptual match between retrieval and encoding. To counter this concern, we derived surrogate reinstatement values for each AR trial by correlating a given AR trial's retrieval patterns with the encoding patterns of all other AR trials where exactly the same target association was presented. These surrogate trials were then averaged to a single AR surrogate

**Table 2.** Behavioural results, separated by colour and scene blocks. **a.** Average (and SEM) proportion of 'plausible' and 'implausible' responses during encoding. **b.** Average (and SEM) proportional associative memory performance for old item hits.

| a. encoding responses | Colour | Scene |
| --- | --- | --- |
| "plausible" | 0.50 (0.03) | 0.46 (0.05) |
| "implausible" | 0.47 (0.03) | 0.50 (0.05) |
| invalid | 0.04 (0.02) | 0.04 (0.02) |
| **b. memory performance** | **Colour** | **Scene** |
| association correct | 0.51 (0.06) | 0.52 (0.05) |
| "don't know" association | 0.32 (0.08) | 0.31 (0.07) |
| association incorrect | 0.16 (0.03) | 0.17 (0.03) |

reinstatement map for each participant. Note that this procedure not only controls for a potential increase in perceptual/attentional reinstatement for AR vs. IR, but it further ensures event-specificity by factoring out shared similarities with all other AR trials where the same colour or scene was recollected.

We then considered retrieval time points to show pattern completion during recollection if there was greater reinstatement (i) for AR vs. IR and (ii) for AR vs. AR surrogates. For each of these comparisons, correlation values were Fisher-z transformed and contrasted via paired-samples t tests. A cluster-based randomization method was used to control for multiple comparisons (each contrast thresholded at P<0.05, corrected; Materials and methods). Results are shown in *Figure 2*. *Figure 2A* depicts the reinstatement map for AR trials. A pronounced cluster of pattern correlations between retrieval and encoding emerged, with stronger effects to the right of the diagonal, indicating that earlier encoding representations tend to get reinstated later during retrieval. Note also that although reinstatement reached its maximum relatively late during retrieval (~1–1.5 s), the effect began to unfold already at ~0.5 s post stimulus onset. The corresponding reinstatement maps for IR trials and for AR surrogate data are presented in *Figure 2B*, showing diminished reinstatement during those conditions compared to AR. *Figure 2C* shows the results of the conjunction of AR > IR and AR > AR surrogates. Surviving this stringent conjunction is an extended cluster where encoding patterns from ~0.5–1 s are reinstated from ~1–1.5 s at retrieval. *Figure 2D* shows the condition-wise reinstatement values underlying the significant conjunction, along with the surrogate data for IR for comparison.

## Hippocampal pattern completion and ongoing oscillatory dynamics

Previous work using intracranial EEG recordings has revealed high-frequency gamma power (~45–95 Hz) increases in the hippocampus not only during successful encoding (*Sederberg et al., 2007b*), but also during subsequent free recall (*Burke et al., 2014*). Our current data allowed us to go one step further and assess whether particular power in- or decreases in the hippocampus are directly linked to our electrophysiological measure of pattern completion. In a first step, we contrasted the time-frequency maps for AR and IR retrieval trials across participants. In particular, participant- and condition-specific time-frequency maps were baseline corrected with respect to a .5 s baseline window and contrasted for AR vs. IR via paired-samples t-tests, again using a non-parametric cluster correction method to account for multiple comparisons. As shown in *Figure 3A*, results revealed two clusters that showed differential effects for AR vs. IR: A relative gamma power (~50–90 Hz) increase for AR from ~0.5 to 1.3 s, followed by a relative alpha power (~8–12 Hz) increase for IR from ~1 to 2 s. The unthresholded difference map is shown in *Figure 3—figure supplement 1*. In *Figure 3B, we* show the power time courses of the significant clusters to better reveal the in- or decreases compared to the prestimulus baseline. The results demonstrate a marked increase in gamma power for AR vs IR relative to their respective baseline windows, whereas the alpha cluster reflects a later increase in alpha power for IR trials but not for AR trials. Interestingly, we found that gamma power from ~0.5 to 1.3 s correlated negatively with alpha power from ~1 to 2 s both on a trial-by-trial level and across participants (*Figure 3—figure supplement 2*).

To ensure that these effects are not driven by condition differences in the pre-stimulus baseline period, we omitted baseline correction and log transformed the raw power data instead (*Cohen, 2014*) (*Figure 3—figure supplement 3*). We then compared AR vs. IR power both in the pre-stimulus baseline window (−0.5 to 0 s) and in the post-stimulus window in which we observed the effects above (0.5 to 1.3 s for gamma and 1 to 2 s for alpha). First, a significant window x condition interaction for gamma power ($F(1,10) = 22.15$, $P = 0.001$) reflected a significant increase for AR vs. IR in the post-stimulus window ($t(10) = 2.65$, $P = 0.024$) but not in the pre-stimulus baseline window ($t(10) = 1.72$, $P>0.1$). Likewise, a significant window x condition interaction for alpha power ($F(1,10) = 32.62$, $P<0.001$) reflected a significant increase for IR vs. AR in the post-stimulus window ($t(10) = 4.84$, $P = 0.001$) but not in the pre-stimulus baseline window ($t(10) = 1.38$, $P>0.1$) and a significant increase from baseline for IR ($t(10) = 2.97$, $P = 0.014$) but not for AR ($t(10) = 0.85$, $P = 0.415$). These results confirm that condition differences during retrieval unfolded after stimulus onset in our paradigm.

Although the timing of the post-stimulus gamma and alpha differences for AR vs. IR (~0.5 to 2 s) overlaps with the time window showing pattern completion for AR trials (1 to 1.5 s), we next sought to establish a stronger link between the power effects and the reinstatement effect observed above.

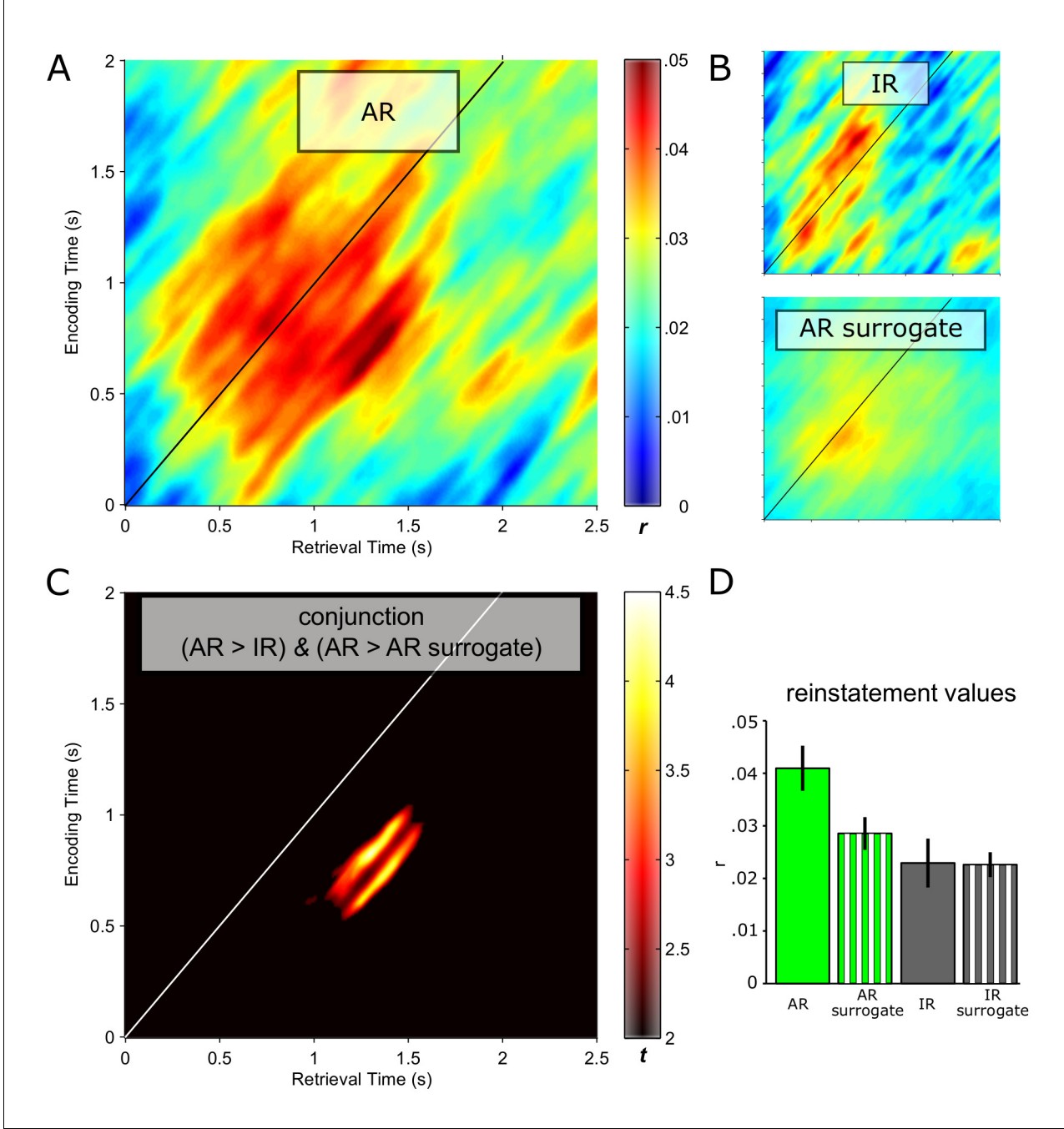

**Figure 2.** Pattern completion in the hippocampus during recollection. Reinstatement maps based on sliding encoding-retrieval pattern correlations are shown for successful associative recognition (AR; (A)), for non-associative item recognition (IR; B *top*) and for AR surrogate data (B, *bottom*). X/Y axes and colour range are identical across panels. (C) Conjunction (minimum t statistic) of significant pairwise comparisons of AR vs. IR and AR vs. AR surrogates, each comparison thresholded at P<0.05 (cluster corrected). The diagonals in A–C highlight same time points at encoding and retrieval. Results show that encoding patterns from ~0.5 to 1 s are reliably reinstated from ~1 to 1.5 s during successful AR. (D) Average (± SEM) reinstatement values of 0.5 to 1 s encoding patterns at 1 to 1.5 s during retrieval are plotted for AR, IR and their respective surrogates for illustration purposes.

The following figure supplement is available for figure 2:

**Figure supplement 1.** Contributions of different frequency bands to reinstatement.

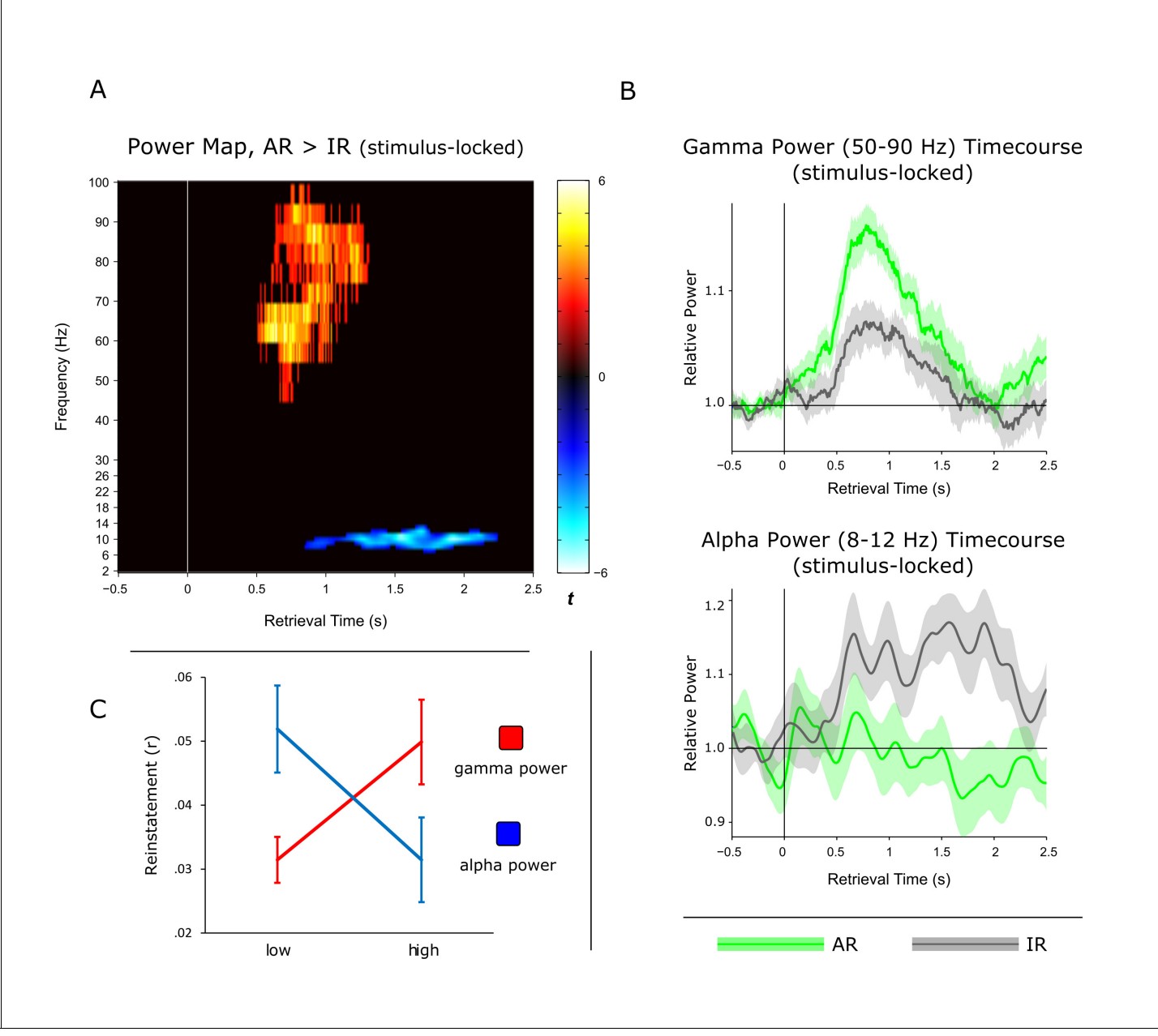

**Figure 3.** Power dynamics during recollection. (**A**) Time-frequency contrast map comparing successful associative recognition (AR) with non-associative item recognition (IR), revealing a cluster of increased power for AR in the gamma band (50–90 Hz) followed by a cluster of increased power for IR in the alpha band (8–12 Hz). Map is thresholded at P<0.05 (corrected), with the unthresholded contrast map shown in *Figure 3—figure supplement 1*. (**B**) The time courses of AR and IR in the resulting frequency bands for illustration of relative power in- and decreases. (**C**) Link between gamma power increases/alpha power decreases during AR and pattern reinstatement. Median-split of power values reveals greater reinstatement during AR trials when gamma power is high and alpha power is low.

The following figure supplements are available for figure 3:

**Figure supplement 1.** Unthresholded time-frequency representation of the contrast associative recognition (AR) > non-associative item recognition (IR).

**Figure supplement 2.** Earlier gamma power and later alpha power are negatively correlated.

**Figure supplement 3.** No condition differences in the baseline period.

*Figure 3 continued on next page*

*Figure 3 continued*

**Figure supplement 4.** Encoding data.

We therefore asked whether power fluctuations in the gamma and alpha bands would correlate with fluctuations in pattern reinstatement across AR trials. To address this question, we divided AR trials into those with relatively high power values and those with relatively low power values (median split) and compared the reinstatement values in the resulting sub-categories. Values were obtained from the time-frequency clusters emerging from the previous analyses, i.e. 50–90 Hz power from 0.5 to 1.3 s for the gamma effect, 8–12 Hz power from 1 to 2 s for the alpha effect, and 0.5 to 1 s encoding time to 1 to 1.5 s retrieval time for the reinstatement effect. Indeed, results revealed a significant interaction of relative power (high, low) x frequency band (gamma, alpha) on reinstatement values (F (1,10) = 12.16, P = 0.006) (*Figure 3C*). Follow-up pairwise t-tests revealed that while AR trials with greater gamma power yielded greater reinstatement than AR trials with lower gamma power (0.050 vs. 031; t(10) = 2.82, P = 0.018), alpha power showed a trend in the opposite direction (0.031 vs 0.052; t(10) = 1.95, P = 0.079). These results point to a functional link between power fluctuations in the gamma band and increases in hippocampal pattern completion during recollection.

Does the gamma power increase during AR trials mean that the critical pattern driving reinstatement is contained in that frequency band, or does reinstatement rely on the representational patterns across a larger frequency range? It is important to note that the surrogate analysis already controls for a potential signal-to-noise confound, as the same trials were used to create the surrogates (thus preserving the time-frequency profiles), but with different assignments of encoding to retrieval trials. Nevertheless, we sought to explore whether any particular frequency range might drive reinstatement in our data. To this end, we conducted the same reinstatement analyses after systematically excluding a particular band (δ, θ, α, β, γ1, γ2) from the time-frequency representations, as well as after including only that frequency band. Results (*Figure 2—figure supplement 1*) showed that while the reinstatement effects prevailed when excluding any particular frequency band, no single band alone was sufficient to drive reinstatement across all participants and trials, suggesting that event-specific hippocampal reinstatement capitalizes on the rich information profile carried by a wider range of frequencies.

## Hippocampal pattern completion, ongoing oscillatory dynamics and recollection

As mentioned above, AR and IR trials had different response latencies, likely to reflect prolonged memory search processes during IR trials. To account for these different response latencies when comparing conditions and to more directly link the pattern completion and oscillatory effects above to the mnemonic outcome of recollection, we repeated the previous analyses after response-locking (instead of stimulus-locking) the data. In particular, we realigned the data with respect to the button press indicating successful or unsuccessful associative recognition and extracted 1 s before and. 1 s after the response. First, for the reinstatement analysis, we again found a cluster showing significant effects for both AR > IR and AR > AR surrogate. Again, the ~0.5–1 s encoding pattern was reinstated during AR trials, with a maximum between −0.8 and −0.2 s prior to the behavioural response (*Figure 4A*). Critically, the response-locked power time courses – particularly the gamma power time course – showed a striking temporal overlap with the response-locked reinstatement time course, both showing their maximum effect size ~0.5 s prior to the memory response.

## Discussion

### Pattern completion in the human hippocampus

The notion of pattern completion, i.e. the representational reinstatement of a memory trace upon receiving a partial cue, is inherent in computational and theoretical accounts of episodic memory (*Marr, 1971*; *McClelland et al., 1995*; *Rolls, 2016*). We propose that in order to qualify as evidence for pattern completion in functional data, at least the following three criteria would need to be met: First, if a link to episodic memory is to be made, pattern completion should be evident after single-

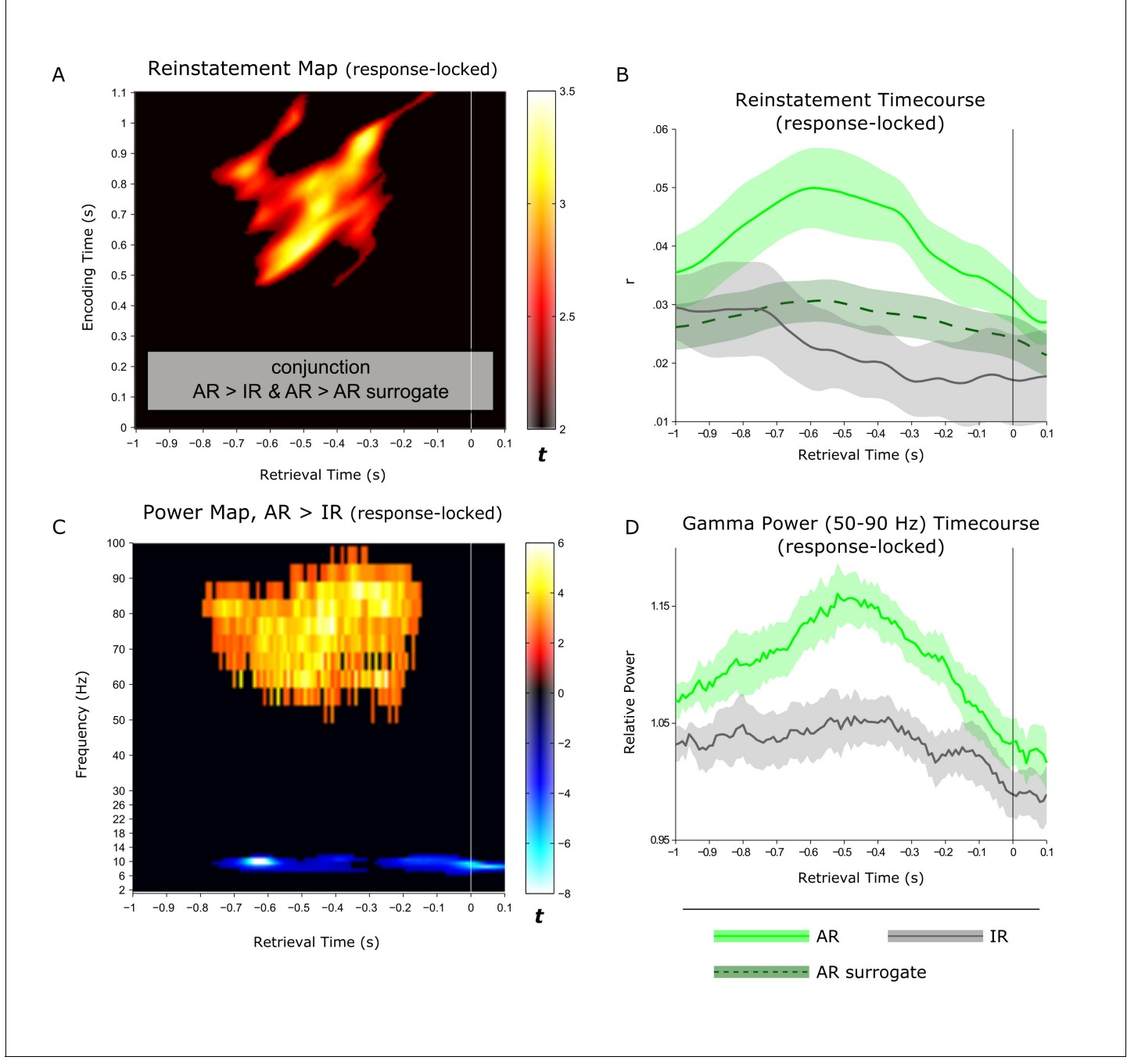

**Figure 4.** Linking hippocampal reinstatement and power dynamics to behaviour. (**A**) Reinstatement map from −1 s to +0.1 s relative to the behavioural response, revealing a significant cluster of ~0.5-1 s encoding patterns being reinstated from ~−0.8 to −0.2 s prior to the button press. (**B**) For visualization, average (± SEM) time-course data are shown for reinstatement of the ~0.5 to 1 s encoding patterns for associative recognition (AR), non-associative item recognition (IR) and AR surrogate data. (**C**) Power contrast map of AR > IR from −1 s to +0.1 s relative to the behavioural response, revealing a significant gamma power (50–90 Hz) cluster from ~−0.8 to −0.2 s during retrieval and a more sustained, narrow-band alpha power (8–12 Hz) cluster. (**D**) For visualization, average (± SEM) power time-course data are shown for associative recognition (AR) and non-associative item recognition (IR).

The following figure supplements are available for figure 4:

**Figure supplement 1.** Response-locked reinstatement map with modified analytical settings.

**Figure supplement 2.** Selectivity of hippocampal gamma oscillations to recollection.

shot learning. Second, to assert the relevance for behaviour, pattern completion should be evident for successful compared to unsuccessful memory performance. Finally, the notion that unique, event-specific representations are reinstated has to be corroborated – in cases where episodic elements overlap across trials (as is the case in our current paradigm) - by controlling for all other instances in which the same episodic elements are also recollected. With these basic criteria in mind, direct empirical evidence for pattern completion in the human hippocampus has thus far been lacking.

For example, recent fMRI studies have shown that overlearned encoding stimuli can be decoded in the hippocampus during recall attempts of those stimuli (*Chadwick et al., 2010*; *Mack and Preston, 2016*). However, hippocampal reinstatement after single exposure and its relation to successful vs. unsuccessful recollection has not been investigated in those studies. Similarly, employing single unit recordings in humans, the same cells in the hippocampus that were activated during the initial encounter (e.g., seeing a video-clip of the Simpsons) were found to re-activate during the subsequent recall of that encounter (*Gelbard-Sagiv et al., 2008*). But again, that approach doesn't allow distinguishing categorical/semantic responses from event-specific/episodic responses – learning stimuli were shown repeatedly and a particular cell may be equally responsive when a different exemplar of that category (e.g., a different clip of the Simpsons) is encountered or recalled. Finally, another set of recent fMRI studies did show evidence for trial-unique/event-specific reinstatement linked to recollection, but only in category-specific neocortical modules and not in the hippocampus (*Ritchey et al., 2012*; *Staresina et al., 2012b*; *Tompary et al., 2016*). Interestingly, in those studies, despite not exhibiting reinstatement itself, the hippocampus has been found to co-vary with reinstatement in cortical regions in terms of BOLD activation levels (see also [*Gordon et al., 2014*]). While consistent with a role of the hippocampus in orchestrating cortical reinstatement (*Staresina et al., 2013a*; *Teyler and DiScenna, 1986*), there must be – despite sparse coding and possible remapping of cells (*Colgin et al., 2008*; *Quiroga et al., 2008*) - some representational overlap between an event's retrieval pattern and its designated encoding pattern in the hippocampus in order to ignite the recollection process. Here we used time-frequency analyses of direct hippocampal recordings to provide evidence for the hypothesized pattern completion processes in the hippocampus. In an episodic memory paradigm (*Figure 1*), we found that reinstatement was greater during successful associative recognition (AR) than during non-associative item recognition (IR) and co-terminated with the behavioural memory response (*Figures 2* and *4A–B*). Note that the comparison with IR trials also rules out that reinstatement during AR merely reflects seeing the same noun on the screen, which would hold for both AR and IR. Likewise, our stringent surrogate analysis controls for conceivably non-specific features shared across AR trials (e.g., greater signal-to-noise ratios), even when the same target association was retrieved. These controls suggest that our results indeed reflect reinstatement of the trial-unique combination of noun, associative detail and potentially other event-specific contextual variables (internal or external and akin to 'holistic pattern completion processes' identified by Horner and colleagues [*Horner et al., 2015*; *Horner and Burgess, 2014*]).

In the stimulus-locked analysis (*Figure 2*), the peak effect entailed ~0.5–1 s encoding patterns being reinstated from 1–1.5 s at retrieval. While also showing that earlier encoding patterns are reinstated later during retrieval, the corresponding time windows were considerably earlier in a recent study assessing memory reinstatement via MEG (where encoding representations around 180 ms were reinstated at ~500 ms at retrieval; [*Jafarpour et al., 2014*]). However, that study focused on classifying category-level patterns (faces vs. scenes, with faces known to elicit an early face-specific response at ~170 ms [*Bentin et al., 1996*]) and did not relate reinstatement to successful vs. unsuccessful retrieval. More importantly, the response-locked analysis of our data (*Figure 4A–B*) shows that reinstatement reaches its maximum ~0.5 s prior to the behavioural response, which is in good agreement with a recent report showing the peak cortical reinstatement during paired-associate word recall in the same time window via electrocorticography (*Yaffe et al., 2014*).

## The role of hippocampal gamma power in recollection

Thus far, the frequency band most closely linked to episodic memory processes in the human hippocampus has been high gamma (*Hanslmayr et al., 2016*), as evidenced for instance by gamma power increases during both successful encoding and recall in a word-list learning paradigm (*Lega et al., 2012*; *Sederberg et al., 2007a*, *2007b*). Here, we were able to show that hippocampal gamma power not only distinguishes successful from unsuccessful associative recognition, but also - within

the AR condition - trials of higher vs. lower levels of reinstatement (*Figure 3C*), extending the link between hippocampal gamma power and episodic memory from solely behavioural to neurophysiological measures of recollection. That said, a recent iEEG study challenges the notion that hippocampal gamma power selectively supports episodic recollection (*Merkow et al., 2015*). After identifying increased gamma power for word recognition hits relative to misses (in a time window overlapping with our current gamma effects), the study set out to assess whether the hippocampal gamma power increases are specific to recollection-based recognition or also distinguish different levels of familiarity-based recognition. In the absence of an associative memory test or confidence judgments, response latencies during recognition were used to construct receiver operating characteristics (ROCs), commonly used to capture the contributions of familiarity vs. recollection to recognition memory (*Yonelinas, 1994*). Results suggested that hippocampal gamma power not only supported recollection, but also scaled with different levels of familiarity-based recognition. In our current paradigm, we were able to address this issue further by comparing hippocampal gamma power not only for AR vs. IR trials, but also for forgotten items (misses; M) as well as correctly identified new items (correct rejections; CR). The level of familiarity should be greater for IR than for M, perhaps with a further decrease from M to CR (reflecting residual familiarity for old items incorrectly classified as new relative to novel items). Correspondingly, one would expect a pattern of AR > IR > M >= CR for a signal that scales with familiarity-based recognition. Contrary to this prediction and as shown in *Figure 4—figure supplement 2*, the increase in gamma power prior to the behavioural response was highly specific to AR trials and did not distinguish further between IR, M and CR trials. To the extent that our AR condition is particularly sensitive to recollection, this finding points, at least in the current paradigm, to a selective role of hippocampal gamma power in recollection-based associative recognition.

How may the increase of hippocampal gamma power during associative recognition be understood in terms of neural mechanisms? A straightforward interpretation would be that this increase reflects selective firing rate increases in neurons representing the retrieved information. Indeed, using single-unit recordings in humans, such selective firing rate increases during memory retrieval were reported for those neurons that previously responded during encoding (*Gelbard-Sagiv et al., 2008*; *Miller et al., 2013*). However, in light of other single-unit studies, the idea of selective firing rate increases may be oversimplified. For instance, in terms of overall responses of human hippocampal neurons, only ~50% of neurons were found to respond with a firing rate increase at the second stimulus presentation compared to the first presentation in a continuous recognition task. In fact, around twice as many hippocampal neurons responded to previously seen stimuli with a firing rate decrease (*Viskontas et al., 2006*). A very similar result has been reported for recordings from macaque hippocampus (*Jutras and Buffalo, 2010*), with almost three times the number of neurons responding with a firing rate decrease to repeated stimuli than with an increase. Together, these findings raise the possibility that hippocampal inhibition plays a major role in episodic memory retrieval. Indeed, inhibitory hippocampal interneuron networks, oscillating at gamma frequencies, control the firing of pyramidal cells and restrict their action potentials to specific time windows by cholinergic inputs (*Bartos et al., 2007*; *Whittington et al., 1995*). Hence, increased gamma power during memory retrieval may first and foremost reflect the temporal alignment (synchronization) of target pyramidal cells, which may co-occur with a net firing rate decrease (e.g., [*Axmacher et al., 2008*]). Accordingly, one tentative interpretation is that the observed gamma power increase for AR vs. IR trials in our study may correspond to a selective recruitment mechanism, prioritizing neurons that represent the target association whilst de-prioritizing neurons representing non-relevant information.

## Inverse relationship of hippocampal gamma and alpha power

Another intriguing finding emerging from our time-frequency analysis was a relative reduction of alpha power during AR compared to IR from ~1–2 s after stimulus onset. Alpha power decreases during successful recognition have been observed in a number of scalp EEG/MEG and electrocorticography studies (for review, see [*Hanslmayr et al., 2012*]). In the hippocampus, fairly broadband low-frequency power decreases (encompassing theta, alpha and beta frequencies) in the hippocampus were found to support successful encoding (*Fell et al., 2011*; *Sederberg et al., 2007b*), but hippocampal low-frequency power decreases during successful retrieval are only rarely reported (*Lega et al., 2012*).

Alpha-band oscillations have traditionally been linked to idling states or the suppression of activity within particular functional networks in sensory cortices (*Berger, 1929*; *Jensen et al., 2012*; *Klimesch, 2012*). Applied to our current results, the relative increase of alpha power for IR (*Figure 3*) may hence reflect disengagement of hippocampal retrieval operations upon failure to surpass a certain recollection threshold. Another recent account proposes that neural desynchronization in alpha/beta bands (expressed in power decreases) reflects an increase in sensory information content (*Hanslmayr et al., 2016*). In that sense, the relative decrease for AR may reflect an increase in information (i.e. the target association) resulting from effective retrieval operations. Accordingly, one would expect a direct link between earlier hippocampal gamma power increases – potentially reflecting the retrieval operations (see above) – and later alpha power decreases (potentially reflecting the amount of information available as a result of those retrieval operations). Indeed, in an additional analysis, we found that gamma power from ~0.5 to 1.3 s correlated negatively with alpha power from ~1 to 2 s both on a trial-by-trial level and across participants (*Figure 3—figure supplement 2*). One tentative interpretation is therefore towards a direct relationship between (i) associative retrieval processes as reflected by the power in the gamma band and (ii) the amount of mnemonic information available as inversely reflected by the power in the alpha band. Pushing this idea one step further, if hippocampal alpha power is indeed related to information content, one might expect alpha differences between AR and IR to emerge earlier during encoding, in line with the notion that relatively unsuccessful encoding results from impoverished levels of incoming information. Interestingly, when contrasting AR with IR trials during encoding, the same alpha band cluster (8–12 Hz) emerged to show greater power for IR relative to AR, but critically with a markedly earlier timing of this effect than during retrieval (~0.2 to 0.9 s during encoding vs. ~1 to 2 s during retrieval; *Figure 3—figure supplement 3*). A stringent test for the hypothesized functional dissociation of gamma vs. alpha effects would be to simultaneously record from category-selective cortical regions and see whether the low frequency power decreases during encoding and retrieval are also observable in those cortical regions (presumably coding for the sensory information content), while the gamma power increases might remain specific to the hippocampus.

## Conclusion

Using direct intracranial recordings during an associative memory paradigm, we provide empirical evidence for a pattern completion mechanism in the human hippocampus that reinstates event-specific encoding patterns during successful recollection. Results further link reinstatement to fluctuations in gamma power, hypothesized to coordinate the selection of target-relevant neurons. Finally, time-shifted hippocampal alpha power showed an inverse relationship to gamma power, potentially reflecting the amount of information relayed to and from the hippocampus during memory formation and retrieval.

# Materials and methods

## Participants and recordings

Intracranial EEG (iEEG) was recorded from patients suffering from pharmaco-resistant epilepsy at the Department of Epileptology, University of Bonn. Depth electrodes were implanted stereotactically, either via the occipital lobe along the longitudinal axis of the hippocampus or laterally via the temporal lobe, during presurgical evaluation (the seizure onset zone could not be precisely determined with noninvasive methods). Depth electroencephalograms were referenced to linked mastoids and recorded with a sampling rate of 1 kHz (bandpass filter: 0.01 Hz (6 dB per octave) to 300 Hz (12 dB per octave)). All patients received anticonvulsive medication (plasma levels within the therapeutic range). Informed consent for the iEEG recordings and the use of the data for research purposes was obtained from all patients. The study was approved by the ethics committee of the Medical Faculty of the University of Bonn. Complimentary analyses from a sub-sample of 6 patients participating in the current experimental paradigm have been reported previously (*Staresina et al., 2012a*, *2013b*).

A total of 15 patients participated in the study, out of which 4 were excluded from subsequent analyses. In three of these patients, clinical monitoring revealed epileptogenic activity in both hippocampi, and one patient was a non-native German speaker and had difficulties understanding the stimulus material. Of the remaining 11 participants, 6 were female and 5 were male. Mean age was

34 years (range: 23–51). Clinical evaluation revealed a unilateral epileptic focus in the left hemisphere of 8 patients (7 in the hippocampus, 1 in the anterior temporal lobe) and in the right hemisphere hippocampus of 2 patients. Only data from the hemisphere contralateral to the seizure onset zone were included. No hippocampal focus was diagnosed in 1 patient, and left hemisphere data were used based on the selection criteria described below. Thus, our sample consisted of 8 right hemisphere and 3 left hemisphere hippocampal datasets.

## Experimental procedures

The experiment was conducted in a sound-attenuated room, with the participant sitting upright in a comfortable chair. A laptop computer, used for stimulus presentation, was positioned on a table at a ~50-cm distance. The experimental paradigm is schematized in *Figure 1*. Each experimental run contained an encoding phase, a 1-min distracter phase and a retrieval phase. During encoding, participants were presented with a German noun paired during colour runs with the colour blue or red, and during scene runs with the image of an indoor or an outdoor scene (office or nature). Colour and scene runs alternated, with the assignment of the first run to colour or scene rotated across participants. The use of colours vs. scenes was initially intended to allow investigation of differential effects across MTL cortical regions (in case a patient would have sufficient electrode coverage), but for the current purposes, we did not further differentiate across these types of associations (note also that behaviour was matched between colour and scene blocks; see *Table 2*). The encoding task was to vividly imagine the referent of the noun in the given colour/scene and to rate the plausibility of that image as plausible or implausible. Participants were given 3 s to make their plausibility judgment. Each trial was preceded by a jittered intertrial interval (700–1300 ms, mean = 1000 ms) during which a fixation cross was shown in the centre of the screen. Trials terminated with the participant's button press. During retrieval, participants were presented with 75 trials including the 50 previously seen words along with 25 novel words. The task was to indicate, with a single button press, whether the word was new, whether it was old but the target association could not be retrieved ('?' responses), or whether the word was old and the target association was also remembered. Responses were given in a self-paced manner, with an upper time limit of 5 s. Again, each trial was terminated with the button press and was preceded/followed by a jittered intertrial interval (700–1300 ms, mean = 1000 ms) showing a fixation cross. Each run lasted ~9 min. Eight participants completed all six runs and three participants completed five runs.

## Electrode selection

For group-level analyses, we selected one hippocampal depth electrode contact per participant based on anatomical and functional criteria. Anterior hippocampal electrodes were favoured because associative memory effects have been reliably observed in this region (for review, see [*Davachi, 2006*]) and recent theoretical accounts postulate a specific role of the anterior hippocampus in pattern completion (whereas the posterior hippocampus might be biased towards pattern separation) (*Poppenk et al., 2013*). Furthermore, anterior hippocampal contacts showed reliable ERP associative recognition effects in a subset of the current sample (*Staresina et al., 2012a*). For initial selection purposes, signal quality was assessed for each channel in terms of artefactual raw trials, where artefacts were defined as time points in which both absolute amplitude and gradients (i.e. the difference between two adjacent time points) exceeded the median plus 3 inter quartile ranges across −1 to +3 s around stimulus onset. This procedure proved sensitive to detecting epileptogenic activation. The lateral implantation scheme (n=4) typically includes depth electrodes with two closely spaced contacts in the anterior hippocampus, and provided that both contacts were located within the hippocampus, the contact with higher signal quality was selected. The longitudinal implantation scheme (n=7) typically contains depth electrodes with 10 evenly spaced contacts, with the anterior ~2–4 contacts located in peri/entorhinal cortex and the remaining ~6–8 contacts spanning anterior to posterior hippocampus (see *Figure 1B* for an example). To complement the anatomical demarcation of rhinal cortex, anterior hippocampus and posterior hippocampus based on the post-implantation MRI, we calculated the pairwise channel coherence (2–10 Hz) to reveal functional transitions from rhinal cortex to anterior hippocampus and from anterior hippocampus to posterior hippocampus ([*Mormann et al., 2008*]; for more details, see [*Staresina et al., 2012a*]). Again, if multiple channels were anatomically located in the anterior hippocampus and were clustered based on the

pairwise channel coherences, the channel with the highest signal quality was selected. In one patient, the delineation between rhinal cortex and anterior hippocampus was less clear, and to avoid ambiguities about rhinal vs. hippocampal signal generators, we selected a contact in the posterior hippocampus for that patient. Results remained unchanged, however, when excluding that patient from the analyses.

To visualize the selected contacts across our sample, we normalized each participant's post-implantation MRI along with their co-registered pre-implantation MRI to MNI space using SPM8 (http://www.fil.ion.ucl.ac.uk/spm/). Based on visual identification of the contact centres, the average xyz coordinates were ±28, −17 and −18 mm. To facilitate the visualization of contacts across the group, a 5-mm-radius sphere was created around each contact's centre point and overlaid across participants (*Figure 1B*).

## Analyses

Data processing was performed with FieldTrip (*Oostenveld et al., 2010*) and standard MATLAB functions. Artifact rejection was performed on trial epochs from −1 to +3 s time locked to stimulus onset. Prior to manual artefact rejection, an automated procedure was implemented to reject trials in which at least one time point exceeded three interquartile ranges of all trial-specific values in both amplitude and gradient (difference to previous time point). Manual artefact rejection was conducted using FieldTrip's summary plot functions followed by trialwise artefact inspection. Across participants, an average of 12% of all trials (range: 7%–21%) were thus excluded.

After artefact rejection, frequency decomposition of the data was achieved via Fourier analysis based on sliding time windows (moving forward in 10-ms increments). The settings were optimized for two frequency ranges. For a lower frequency range (2–29 Hz, 1-Hz steps), the window length was set to five cycles of a given frequency (for example, 500 ms for 10 Hz; 250 ms for 20 Hz), and the windowed data segments were multiplied with a Hanning taper before Fourier analysis. For higher frequencies (30–100 Hz, 5-Hz steps), we applied multitapering, using a fixed window length of 400 ms and seven orthogonal Slepian tapers (resulting in spectral smoothing of ~±10Hz) (this approach was adopted from [*Jokisch and Jensen, 2007*]). Note that the same pattern of results was observed when shortening the time windows, e.g. to four cycles of the lower frequencies and 200 ms for higher frequencies. The resulting power maps were normalised by dividing over the averaged −0.5 s prestimulus baseline window and subjected to direct comparison between conditions of interest.

It deserves explicit mention that we used local time-frequency patterns recorded from a single (anterior) hippocampal electrode to quantify reinstatement. While this approach differs from other studies using spatial patterns across neocortical channels recorded via scalp EEG (*Staudigl et al., 2015*) or ECoG (*Yaffe et al., 2014*), recent rodent studies have demonstrated that – in the hippocampus – memory reinstatement can be observed in highly localized cell assemblies (*Liu et al., 2012*; *Tayler et al., 2013*). Moreover, as comprehensively reviewed by Buzsaki and colleagues (*Buzsáki et al., 2012*), the field potential picked up by a single intracranial macro-electrode reflects contributions from all active cellular processes within the underlying volume of brain tissue (including synaptic activity, action potentials, fluctuations in glia etc.). As amplitude and frequency of the recorded field potential depend on the proportional contribution of these multiple sources, the time-frequency-decomposed signal might be particularly well suited to capture local hippocampal pattern completion processes in our study (i.e. firing patterns of distributed neurons bias the measured field potentials differentially; see also [*Agarwal et al., 2014*], although that study used even finer-grained micro-electrode field recordings from the rodent hippocampus).

For statistical comparisons of reinstatement maps or power maps, we used a non-parametric cluster-based permutation procedure implemented in FieldTrip (*Maris and Oostenveld, 2007*). The alpha level was set to 5% across all analyses, and parametric t tests were always two-tailed.

## Robustness of reinstatement results to different analysis settings

A recent study used multi-channel electrocorticography to assess reinstatement by correlating frequency x channel patterns after averaging power values across 500 ms time bins (*Yaffe et al., 2014*). Here, as we focused our analysis on data from a single hippocampal contact, we increased the feature space by including frequency patterns over time (41 time points). However, to ensure that our results do not hinge on a particular set of parameters, we repeated the main reinstatement

analysis with a number of different settings. As before, pattern completion was defined as a significant increase for AR vs. IR and for AR vs. AR surrogates. Results are reported for the encoding/retrieval time windows where effects are described in the main text, i.e. 0.5 to 1 s encoding/1 to 1.5 s retrieval for the stimulus-locked analysis and 0.5 to 1 s encoding/−0.8 to −0.2 s retrieval for the response-locked analysis.

In the first variant, we used only the 43 frequency values at a given time point as the representational pattern, i.e. without concatenating or averaging across multiple time points. All pairwise comparisons (AR vs. IR stimulus-locked, AR vs. AR surrogate stimulus-locked, AR vs. IR response-locked, AR vs. AR surrogate response-locked) were significant (all $t(10) > 2.66$, $P<0.024$). Next, we extended our time window from 400 ms to 500 ms, with the representational pattern thus containing 43 x 51 power values. Again, all four pairwise comparisons were significant (all $t(10) > 2.90$, $P<0.017$). Finally, to match the above-mentioned study (*Yaffe et al., 2014*) as closely as possible, we averaged the power values across those 500 ms, with the representational pattern thus containing 43 power values (one value integrating power across 500 ms at each frequency). As before, all four pairwise comparisons were significant (all $t(10) > 2.37$, $P<0.040$). To facilitate comparison across studies, we also show the reinstatement map for AR trials as well as the statistical conjunction map of AR > IR and AR > AR surrogate resulting from this procedure (*Figure 4—figure supplement 1*; response-locked as in Yaffe et al.), highlighting the similarity in the result patterns between the two studies (note though that Yaffe et al. did not report data from the hippocampus).

## Acknowledgements

This work was supported by a Sir Henry Dale Fellowship to BPS jointly funded by the Wellcome Trust and the Royal Society (107672/Z/15/Z). NA receives funding via the DFG Emmy Noether grant AX82/2, DFG grant AX82/3 and SFB 874. NA and JF are supported by the DFG SFB 1089.

## Additional information

### Funding

| Funder | Grant reference number | Author |
| --- | --- | --- |
| Wellcome Trust | 107672/Z/15/Z | Bernhard Staresina |
| Royal Society | 107672/Z/15/Z | Bernhard Staresina |
| Deutsche Forschungsgemeinschaft | AX82/2 | Nikolai Axmacher |
| Deutsche Forschungsgemeinschaft | AX82/3 | Nikolai Axmacher |
| Deutsche Forschungsgemeinschaft | SFB 874 | Nikolai Axmacher |
| Deutsche Forschungsgemeinschaft | SFB 1089 | Nikolai Axmacher Juergen Fell |

The funders had no role in study design, data collection and interpretation, or the decision to submit the work for publication.

### Author contributions

BPS, Conception and design, Acquisition of data, Analysis and interpretation of data, Drafting or revising the article; SM, MB, OJ, Analysis and interpretation of data, Drafting or revising the article; NA, JF, Conception and design, Analysis and interpretation of data, Drafting or revising the article

### Author ORCIDs

Bernhard P Staresina, http://orcid.org/0000-0002-0558-9745

## Ethics

Human subjects: Informed consent for the iEEG recordings and the use of the data for research purposes was obtained from all patients. The study was approved by the ethics committee of the Medical Faculty of the University of Bonn.

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
