## [Decision Letter]

Thank you for submitting your article "Hippocampal pattern completion is linked to gamma power increases and alpha power decreases during recollection" for consideration by *eLife*. Your article has been reviewed by three peer reviewers, including Jeremy Manning (Reviewer #1), and the evaluation has been overseen by Neil Burgess as the Reviewing Editor and Timothy Behrens as the Senior Editor.

The reviewers have discussed the reviews with one another and the Reviewing Editor has drafted this decision to help you prepare a revised submission.

The referees found the study a compelling and novel account of reinstatement of encoded information during recall. The design, focussing on associative memory, showing reinstatement of patterns of oscillatory power corresponding to remembered associative information, was elegant and nicely controlled for non-associative recognition of the cue and for retrieval of the target. In addition, gamma and alpha power increased or decreased respectively between associative retrieval (AR) versus item retrieval (IR) trials. The paper is potentially suitable for publication in *eLife*, but some further work is required.

The two main issues are as follows:

The interpretation of the gamma power increase with AR should be strengthened, given that gamma apparently did not contribute to reinstatement. In this context it would be useful to clarify the relative contribution of different frequency bands to reinstatement, with further reinstatement analyses leaving out different frequencies.

Further discussion of the interpretation of reinstatement of time-varying patterns at a single electrode would be beneficial, specifically as compared to the more commonly used analyses which include a spatial component, i.e. reinstatement of patterns of activity across voxels or across multiple contacts (e.g., Yaffe et al., 2014; see also Staudigl et al., J Neurosci 35: 5373, 2015). A related issue concerns the interpretation of pattern completion in the frequency domain – to what extent does this necessarily relate to the reinstatement activity patterns across neural populations (which is presumably what we are ultimately interested in)?

More minor issues include the following:

The term pattern completion could be clarified. For some it simply means associative memory (the spreading of activity from the cue to the target), as used here. But it also has a more specific implication of automatic and complete reinstatement of the entire stored pattern, which would incidental internal and external information other than the target in the type of episodic memory discussed here, but might not in semantic memory (see Horner et al., 2015). The paradigm here does not address this more specific meaning.

Further examination and discussion of reinstatement of cue information in IR trails would be of interest, why is it apparently weaker than reinstatement in AR trials?

Were there any stimulus-locked ERP effects here, or is all relevant oscillatory power 'induced' rather than 'evoked'? This might be an important consideration for interpreting the gamma and alpha power findings, and it would be useful to confirm that reinstatement did not reflect correlations between encoding and retrieval in signal amplitude over time.

The idea of 'hippocampal inhibition' deserves further consideration - although repeated presentation of a stimulus often leads to reduced firing rates, this is usually interpreted in terms of item recognition or familiarity rather than associative recall (most obviously Xiang and Brown, Neuropharmacol. 37:657-676, 1998). It is not clear that reduced gamma implies 'inhibition' in the hippocampus, or just that (perhaps unlike neocortex) overall activity is not reflected in gamma, but perhaps in alpha (e.g. in freely moving rodents, the correlations of theta power and firing rate with running speed would produce a correlation between firing rate and theta power).

Reviewer #1:

To study associative recollection, the authors employed a clever associative memory paradigm whereby iEEG participants encoded trial-unique concrete nouns paired with one of four associative details (blue, red, indoor scene, or outdoor scene). During a series of retrieval tests, participants viewed old and new nouns and were asked to identify the nouns as old/new and to recollect the associated details for old items.

The authors examined a single electrode from the anterior hippocampus from each participant. Through a series of careful analyses, they demonstrated that the time-frequency patterns during encoding were reinstated during retrieval. In one set of analyses, they showed that the power spectra recorded following stimulus onset (during encoding) are reinstated most strongly (following the retrieval cue) when participants were also able to recollect the target details associated with the recalled item (AR trials). They ensured they were identifying event-specific reinstatement by comparing the degree of reinstatement during AR trials with the degree of reinstatement during "surrogate" AR trials that shared the same target association. I thought this comparison was especially stringent. Reinstatement was also stronger during AR trials than IR trials (when the participants recalled the items but not the associated details). In a second set of analyses, the authors found that, at retrieval, participants exhibited a greater gamma increase and alpha decrease during AR vs. IR trials. They examined the detailed time courses of these latter effects to demonstrate that reinstatement during IR trials happens over a longer timeline as compared with AR trials.

In general I think the analyses were solid and interesting, and the authors have done their "due diligence" in ruling out potential confounds.

I have two suggestions, both of which I think can be addressed without new analyses or data collection:

The "hippocampal inhibition" story presented in the Discussion section is not directly supported by this study. The wording of that section should be toned down even more to emphasize that the inhibition interpretation is speculative so that there's no confusion about what is specifically shown by this study.

What is a pattern? In the present study, a pattern is a power spectrum at a given moment (or, in some parts of the paper, a frequency-domain pattern extending over time). In the fMRI literature, and even in many of the ECoG studies the authors cite as prior evidence of pattern reinstatement, "patterns" are cast as multi-voxel or multi-electrode responses. It seems clear that the LFP responses the authors are considering reflect population responses, but because those responses can only be measured via a single electrode, it's the fMRI equivalent to considering only a single voxel. I don't think any of this is a problem per se, but I encourage the authors to emphasize more strongly that they are specifically referring to temporal patterns, as opposed to spatial patterns.

*Reviewer #2:*

The authors show that successful associative recognition exudes increased gamma and decreased alpha power in intracranial electrodes in the human hippocampus. Altogether, the study was novel, well-written, and well-argued, with no major flaws. However, I am unsure its level of impact is fit for *eLife* as opposed to a more specialized journal. The paper's impact may depend on the results and interpretation of the major point I raise below.

Major point:

I found this result interesting:

"Using the gamma band frequencies alone did not yield a significant increase for AR vs. IR or AR vs. AR surrogate (both t(10) <.52, P >.622). However, both the differences between AR vs. IR and AR vs. AR surrogate were again significant after excluding the 50-90 Hz frequency range (both t(10) > 2.35, P <.041), suggesting that the reinstated pattern per se might largely be carried by lower frequencies."

However, I think it would strengthen the paper to probe further into which specific frequency bands are indeed relevant for pattern reinstatement. The results would seem to have different meaning if reinstatement were driven by theta, alpha, beta, etc.

Following this point, in the Discussion, the authors say:

"Consistent with this idea that gamma power reflects a facilitating role in episodic reinstatement rather than the firing of the neurons representing the target information, we observed greater gamma power in trials with greater reinstatement, while removal of the gamma band left reinstatement values per se unaffected."

I found the discussion that debunked how previous notions that increased gamma power merely reflected increased neuronal activity enlightening. However, I don't understand what point the authors are trying to make by saying gamma power was higher in trials with greater reinstatement but removing the gamma band leaves reinstatement unaffected. It suggests gamma may merely result from another process that isn't currently shown in the paper. This point may gain clarity if the authors perform the analyses I mentioned above.

*Reviewer #3:*

This study leverages intracranial EEG recordings from human participants to find evidence for reinstatement of trial-specific time-frequency patterns in the hippocampus during successful associative retrieval. Further, the degree of hippocampal reinstatement scales with gamma power increases and alpha power decreases. I appreciated the control analyses, including comparison to the average similarity of all successful retrieval events of the same target association ("AR surrogate"), computing similarity without gamma frequencies, and computing similarity collapsing across time, which strengthen the findings. Overall, the results are convincing evidence for hippocampal reinstatement in the time-frequency domain, though interpretation of reinstatement in the frequency (rather than spatial) domain is needed.

Major comments:

1) Defining a pattern across time/frequency vs. across space: Models of pattern completion generally assume that events are represented by sparse patterns of activity across neurons, and connections formed between these neurons during learning then facilitate activation of the full network of neurons given partial input. Based on this model, the pattern of activity that is "completed" during retrieval would be spatial in nature (i.e., across neurons). However, rather than utilizing the spatial data from channels along the long axis of the hippocampus, only one channel was selected, and the "pattern" of activity evoked by an event was in the temporal and frequency domain (and frequency information alone -collapsing across time - was sufficient to find the reported effects). Why is this the case? The analyses investigating the effects of the gamma power increase on pattern reinstatement were a nice addition (and suggest that the similarity for AR trials is not driven solely by increased signal in the gamma band). However, while reinstatement was still significant within the time window of interest, would the increased similarity for AR > IR and AR > AR surrogate hold with cluster correction when excluding gamma frequencies? If not, it seems misleading to say that the "reinstated pattern per se might be largely carried by lower frequencies." Which frequency bands are contributing to the increased pattern similarity? It seems that the similarity information comes from the frequency bands (rather than a temporal pattern unfolding over time), given that the same pattern of results is found when averaging across time, and it might be informative to re-run analyses excluding certain bands to see whether a given band has a differential contribution to the pattern similarity.

2) Selection of hippocampal channels: Why was only one electrode selected per participant? There were 7 subjects with a longitudinal implantation scheme, allowing for ~6-8 contracts along the long-axis of the hippocampus; given this data, it seems that the patterns could have been calculated across space, which would be more consistent with previous notions of "pattern completion." Secondly, why was the chosen electrode selected in the anterior hippocampus? Given some literature suggesting that posterior hippocampus is involved in retrieval, particularly of spatial information, and given that half of the associates were scenes, why weren't posterior hippocampus electrodes chosen (or at the very least, why were they excluded)?

3) Response time should be reported: It would be helpful to see the response time data. Are there differences in RT between color and scene trials? Is there a difference in RT between "don't know" and "association incorrect" trials?

4) What is the degree of pattern reinstatement for IR trials? Given that approximately 1/3 of the "IR" trials are "association incorrect" trials (during which time participants may be reinstating another event, which could lead to increased dissimilarity relative to the "don't know" trials), it might also be informative to calculate pattern similarity across "don't know" and "association incorrect" trials separately. In addition, looking at the plot of IR reinstatement (Figure 2), it appears that there may be a cluster of significant reinstatement of 1-1.5s encoding at <1s retrieval; is this significant with an IR > IR surrogate contrast?

---

## [Author Response]

*The referees found the study a compelling and novel account of reinstatement of encoded information during recall. The design, focussing on associative memory, showing reinstatement of patterns of oscillatory power corresponding to remembered associative information, was elegant and nicely controlled for non-associative recognition of the cue and for retrieval of the target. In addition, γ and α power increased or decreased respectively between associative retrieval (AR) versus item retrieval (IR) trials. The paper is potentially suitable for publication in eLife, but some further work is required.*

We thank all reviewers and the Reviewing Editor for their positive and constructive feedback. As detailed below, we were able to address all remaining issues and to strengthen the paper with additional analyses. Modifications to the manuscript in response to the reviewers’ suggestions are highlighted in green in the revised manuscript.

*The two main issues are as follows:*

*The interpretation of the gamma power increase with AR should be strengthened, given that gamma apparently did not contribute to reinstatement. In this context it would be useful to clarify the relative contribution of different frequency bands to reinstatement, with further reinstatement analyses leaving out different frequencies.*

*Further discussion of the interpretation of reinstatement of time-varying patterns at a single electrode would be beneficial, specifically as compared to the more commonly used analyses which include a spatial component, i.e. reinstatement of patterns of activity across voxels or across multiple contacts (e.g., Yaffe et al., 2014; see also Staudigl et al., J Neurosci 35: 5373, 2015). A related issue concerns the interpretation of pattern completion in the frequency domain – to what extent does this necessarily relate to the reinstatement activity patterns across neural populations (which is presumably what we are ultimately interested in)?*

We agree that both these points deserve further clarification, and to make our case we’ll start with the second point. First and foremost, we agree that representational patterns ultimately reflect engagement of spatially distributed cell assemblies. Importantly though, we argue that the time-frequency-decomposed signal recorded from a single intracranial macro-electrode is well suited to capture the differential contributions of such distributed cell assemblies. As comprehensively reviewed recently by Buzsaki and colleagues (Nature Reviews Neuroscience, 2012), the field potential picked up by a given recording site reflects contributions from all active cellular processes within the underlying volume of brain tissue (synaptic activity, action potentials, fluctuations in glia etc.). In return, amplitude and frequency of the recorded field potential depend on the proportional contribution of these multiple sources. From a theoretical view, the single-electrode field potential is therefore well suited to capture the local ensemble dynamics representing individual memory episodes (Tayler et al., Current Biology 2013; Liu et al., Nature 2012). Indeed, a recent study was able to show that locally distributed hippocampal field potentials contain sufficient information to decode a rat’s spatial location (Agarwal et al., Science 2014). We now augmented our manuscript with additional discussion of this issue.

On that note, one wouldn’t necessarily expect inclusion of more widespread hippocampal contacts to increase the sensitivity of our measure, as the regional specificity of pattern completion might get somewhat diluted. As detailed below, this is indeed what we found, i.e., including posterior hippocampal contacts did not increase reinstatement levels. It is interesting to note that the mentioned studies by Yaffe et al. (2014) and Staudigl et al. (2015) that relied on multi-contact/electrode correlations (see also Jafarpour et al., 2014) did not record activity directly from the hippocampus. Future work might be able to reveal whether the spatial scale of reinstatement differs between hippocampus and neocortex.

Regarding the first point and from the considerations above, if the contributions of different local cell assemblies are expressed in a complex time-frequency representation, a single frequency band is unlikely to carry all information characterizing a particular event. We initially only excluded the 50-90 Hz band to pre-empt the concern that greater reinstatement for AR might merely reflect greater power and hence signal-to-noise ratios in the gamma range (although this is already accounted for by our surrogate analysis). Following the reviewers’ suggestions, we now systematically and analytically examined the contribution of individual frequencies to our effects. No single frequency band emerged as driving reinstatement across all trials and participants, which is expected if reinstatement capitalizes on the event-specific variability in spatial and spectral pattern distributions. This is now reported in the Results section, with an additional supplemental figure (Figure 2—figure supplement 1).

More minor issues include the following:

The term pattern completion could be clarified. For some it simply means associative memory (the spreading of activity from the cue to the target), as used here. But it also has a more specific implication of automatic and complete reinstatement of the entire stored pattern, which would incidental internal and external information other than the target in the type of episodic memory discussed here, but might not in semantic memory (see Horner et al., 2015). The paradigm here does not address this more specific meaning.

We agree that the term pattern completion deserves more clarification. As correctly pointed out, we use a cue-target associative memory paradigm to assess pattern completion/reinstatement. But we would argue that our findings are quite consistent with the operationalization of *holistic pattern completion* used by Horner et al. (2015)/Horner and Burgess (2014). That is, our contrasts of (i) AR vs. IR and (ii) AR vs. AR surrogates ensure that event-specific reinstatement exceeds (i) cue similarity between encoding and retrieval (ii) target association similarity shared across trials, together suggesting that AR reinstatement additionally reflects some incidental internal and/or external information, idiosyncratic to the current event. We now highlight the conceptual similarity to Horner et al.’s definition in the Discussion section.

*Further examination and discussion of reinstatement of cue information in IR trails would be of interest, why is it apparently weaker than reinstatement in AR trials?*

We assume that the reviewer refers to the apparent difference between AR surrogates vs. IR surrogates shown in Figure 2. The key here is that AR surrogates reflect similarity between *remembering a given trial’s target association* and all other encoding trials where the same *target association* was later remembered. Conversely, IR surrogates reflect similarity between *not* remembering a given trial’s target association and all other encoding trials where the same target association was later *not* remembered. Thus, compared to the IR surrogates, there is still substantial mnemonic similarity in the AR surrogates, carried by recollection of the same target association (which makes them such stringent controls for event specificity in the contrast AR vs. AR surrogates). Accordingly, one would expect diminished similarity between a given AR trial and all other encoding trials in which *the other target association from the same category* is later remembered. In other words, comparing an AR trial where the target association *blue* is remembered with a trial where the target association red was later remembered should yield less similarity than comparing it with another trial where blue was later remembered (c.f., blue car and blue cup in Figure 1). In fact, because no mnemonic details are shared in these same *category/other target* surrogates, their reinstatement levels should not differ from IR trials (which are also characterized by the absence of shared associative mnemonic details from encoding to retrieval).

Indeed and as illustrated below, the level of same *category/other target* AR surrogate similarity did not differ (i) from IR (t(10) = 0.64, P =. 539), (ii) from same *category/same target* IR surrogates (t(10) = 1.40, P =. 192) or (iii) from same *category/other target* IR surrogates (t(10) = 0.99, P =. 345). These results corroborate the notion that hippocampal reinstatement scales with the amount of associative mnemonic overlap between encoding and retrieval trials.10.7554/eLife.17397.015Author Response Image 1.**DOI:**
http://dx.doi.org/10.7554/eLife.17397.015

*Were there any stimulus-locked ERP effects here, or is all relevant oscillatory power 'induced' rather than 'evoked'? This might be an important consideration for interpreting the gamma and alpha power findings, and it would be useful to confirm that reinstatement did not reflect correlations between encoding and retrieval in signal amplitude over time.*

This is a good point. Although the reinstatement effects unfolded past the early time window typically showing more evoked components, we analytically confirmed the reliance of our effects on induced components in two ways. First, we subtracted each participant’s mean ERP (averaged across all trials) from each individual trial prior to deriving the time-frequency power representations (TFRs). Second, we did the same after subtracting the condition-specific ERP (i.e., the ERP averaged across all AR encoding, AR retrieval, IR encoding and IR retrieval trials) prior to deriving the time-frequency power representations for each of those conditions. Both analyses yielded the same results as shown in the main analyses.10.7554/eLife.17397.016Author Response Image 2.**DOI:**
http://dx.doi.org/10.7554/eLife.17397.016

*The idea of 'hippocampal inhibition' deserves further consideration - although repeated presentation of a stimulus often leads to reduced firing rates, this is usually interpreted in terms of item recognition or familiarity rather than associative recall (most obviously Xiang and Brown, Neuropharmacol. 37:657-676, 1998). It is not clear that reduced gamma implies 'inhibition' in the hippocampus, or just that (perhaps unlike neocortex) overall activity is not reflected in gamma, but perhaps in alpha (e.g. in freely moving rodents, the correlations of theta power and firing rate with running speed would produce a correlation between firing rate and theta power).*

We agree with the reviewer that none of our effects reflect ‘inhibition’ as typically observed in reduced firing rates after repeated exposure to a stimulus and often linked to familiarity (Xiang and Brown, 1998). What we refer to as inhibition in the context of interpreting the hippocampal gamma increase for AR trials is the putative role that inhibitory interneurons play in the generation of gamma (e.g., Bartos et al., Nature Reviews Neuroscience, 2007; Wang and Buzsaki, 1996). That is, we speculate that the gamma effect reflects the mechanistic process of selecting relevant neuronal assemblies needed for successful reinstatement. The more successfully such a selection process is deployed, the greater the extent of trial-specific reinstatement, which is what we observe in the correlation between gamma power levels and reinstatement during AR trials. Given the paucity of data on alpha in the human hippocampus, interpretation of the increase for IR vs. AR has to remain speculative. We discuss a potential scenario in which an alpha power increase reflects a decrease in the amount of information processed and/or represented, as so consistently observed in cortical regions (Klimesch, TICS 2012). Albeit speculative, this interpretation fits with the inverse relationship to the preceding gamma and reinstatement effects, such that the relative alpha power increase for IR trials reflects the relative decrease in informational content as a function of less effective recollection/reinstatement

*Reviewer #1:*

*To study associative recollection, the authors employed a clever associative memory paradigm whereby iEEG participants encoded trial-unique concrete nouns paired with one of four associative details (blue, red, indoor scene, or outdoor scene). During a series of retrieval tests, participants viewed old and new nouns and were asked to identify the nouns as old/new and to recollect the associated details for old items.*

*The authors examined a single electrode from the anterior hippocampus from each participant. Through a series of careful analyses, they demonstrated that the time-frequency patterns during encoding were reinstated during retrieval. In one set of analyses, they showed that the power spectra recorded following stimulus onset (during encoding) are reinstated most strongly (following the retrieval cue) when participants were also able to recollect the target details associated with the recalled item (AR trials). They ensured they were identifying event-specific reinstatement by comparing the degree of reinstatement during AR trials with the degree of reinstatement during "surrogate" AR trials that shared the same target association. I thought this comparison was especially stringent. Reinstatement was also stronger during AR trials than IR trials (when the participants recalled the items but not the associated details). In a second set of analyses, the authors found that, at retrieval, participants exhibited a greater gamma increase and alpha decrease during AR vs. IR trials. They examined the detailed time courses of these latter effects to demonstrate that reinstatement during IR trials happens over a longer timeline as compared with AR trials.*

*In general I think the analyses were solid and interesting, and the authors have done their "due diligence" in ruling out potential confounds.*

*I have two suggestions, both of which I think can be addressed without new analyses or data collection:*

*The "hippocampal inhibition" story presented in the Discussion section is not directly supported by this study. The wording of that section should be toned down even more to emphasize that the inhibition interpretation is speculative so that there's no confusion about what is specifically shown by this study.*

We agree with the reviewer and added text to the corresponding paragraph highlighting the speculative nature of our interpretation. In addition (and in response to Reviewer 2’s comment and the ensuing new analyses) we removed the last sentence about the facilitating role of gamma from the Discussion paragraph ‘The role of hippocampal gamma power in recollection’.

*What is a pattern? In the present study, a pattern is a power spectrum at a given moment (or, in some parts of the paper, a frequency-domain pattern extending over time). In the fMRI literature, and even in many of the ECoG studies the authors cite as prior evidence of pattern reinstatement, "patterns" are cast as multi-voxel or multi-electrode responses. It seems clear that the LFP responses the authors are considering reflect population responses, but because those responses can only be measured via a single electrode, it's the fMRI equivalent to considering only a single voxel. I don't think any of this is a problem per se, but I encourage the authors to emphasize more strongly that they are specifically referring to temporal patterns, as opposed to spatial patterns.*

We agree with the reviewer that our definition of ‘pattern’ requires further clarification. As mentioned above, the field potential recorded at a single intracranial electrode reflects the contributions of multiple local generators. Amplitude and frequency of the recorded field potential thus depend on the proportional contribution of each of these multiple sources. By the same token, the time-frequency decomposition of the signal is sensitive to the differential contributions of a number of these underlying sources, which might be just the spatiotemporal resolution needed to detect local hippocampal pattern completion processes.

As such, the recorded signal is arguably considerably more complex and information-rich than a single voxel in an fMRI study (where one typically gets only one scalar per trial as a proxy for ‘activation’). As detailed in response to Reviewer 3, inclusion of additional electrodes did not increase the correlation values.

Following the reviewer’s suggestion, we now add an entire paragraph to the Methods section emphasizing that our analysis capitalizes on the time-frequency patterns of the field potential:

“It deserves explicit mention that we used local time-frequency patterns recorded from a single (anterior) hippocampal electrode to quantify reinstatement. [...] As amplitude and frequency of the recorded field potential depend on the proportional contribution of these multiple sources, the time-frequency-decomposed signal might be particularly well suited to capture local hippocampal pattern completion processes in our study (i.e., firing patterns of distributed neurons bias the measured field potentials differentially; see also Agarwal et al., 2014).”

Reviewer #2:

*The authors show that successful associative recognition exudes increased gamma and decreased alpha power in intracranial electrodes in the human hippocampus. Altogether, the study was novel, well-written, and well-argued, with no major flaws. However, I am unsure its level of impact is fit for eLife as opposed to a more specialized journal. The paper's impact may depend on the results and interpretation of the major point I raise below.*

*Major point:*

*I found this result interesting:*

*"Using the gamma band frequencies alone did not yield a significant increase for AR vs. IR or AR vs. AR surrogate (both t(10) <.52, P >.622). However, both the differences between AR vs. IR and AR vs. AR surrogate were again significant after excluding the 50-90 Hz frequency range (both t(10) > 2.35, P <.041), suggesting that the reinstated pattern per se might largely be carried by lower frequencies."*

*However, I think it would strengthen the paper to probe further into which specific frequency bands are indeed relevant for pattern reinstatement. The results would seem to have different meaning if reinstatement were driven by theta, alpha, beta, etc.*

*Following this point, in the Discussion, the authors say:*

*"Consistent with this idea that gamma power reflects a facilitating role in episodic reinstatement rather than the firing of the neurons representing the target information, we observed greater gamma power in trials with greater reinstatement, while removal of the gamma band left reinstatement values per se unaffected."*

*I found the discussion that debunked how previous notions that increased gamma power merely reflected increased neuronal activity enlightening. However, I don't understand what point the authors are trying to make by saying gamma power was higher in trials with greater reinstatement but removing the gamma band leaves reinstatement unaffected. It suggests gamma may merely result from another process that isn't currently shown in the paper. This point may gain clarity if the authors perform the analyses I mentioned above.*

We thank the reviewer for this insightful comment. We followed his/her advice and examined the contribution of different frequencies in a systematic and analytical fashion (see also our response to Reviewer 3). First, we re-conducted the analyses (i) relying on all frequency bands but one, systematically excluding one band after the other, and (ii) relying on one frequency band only. We selected the following bands: delta (2-4 Hz), theta (4-8 Hz), alpha (8-12 Hz), beta (12-30 Hz) gamma 1 (30-50 Hz) and gamma 2 (50-100 Hz). We then compared both the reinstatement values for AR trials as well as the conjoint effect of AR > AR surrogates & AR > IR to see whether a particular band is necessary (*exclusion* analysis) or sufficient (*band only* analysis) to drive the observed results. Second, to capture the contribution of each frequency (without binning frequencies into predefined bands), we analytically decomposed the correlation coefficient into the summed product of standard scores. This product would normally be summed across both the 41 time bins and the 43 frequency bins to yield the correlation coefficient, but by leaving the frequency component un-summed one can appreciate the relative contribution of each frequency to the final correlation (‘correlation weights’).

In both analyses, results suggest that no single frequency band would be necessary or sufficient to carry the reinstatement effect across participants. And although null effects have to be interpreted with caution, this result aligns well with our intuition that event-specific reinstatement capitalizes on trial-by-trial variability in spectral contributions, drawing on much larger time-frequency information than could be contained in a single frequency/band.

We now include this analysis in the supplemental material (Figure 2—figure supplement 1; see below) and refer to it after asking whether the gamma power increase for AR trials might account for reinstatement effects in the Results section. In addition, we emphasize that the surrogate analysis actually already controls for this potential confound, as AR surrogates have exactly the same spectral properties as the real AR bin.

Finally, with regard to gamma power in particular, we would indeed argue that it reflects a process that is different from the phenomenon of event reinstatement per se. First, from an analytical point, the condition-wise power comparisons yielding the gamma increase and alpha decrease are very different from the reinstatement analysis: the gamma effect of AR > IR reflects a consistent increase in gamma power that is reliable across trials and participants. In our interpretation, we speculate that this gamma effect reflects engagement of inhibitory interneurons deployed to select the neural contributors most informative/relevant for successful reinstatement of a particular event.

Conversely, the reinstatement analysis is a multivariate approach that simultaneously incorporates a range of information (including frequency and time). It is flexible as to whether power at a specific frequency increases in one trial but decreases in another trial and whether it varies across participants, as long as it shows a similar pattern during a particular trial’s encoding and retrieval. Another way to conceptualize this difference is that the AR gamma power increase reflects a *process* (thought to reflect coordination of effective retrieval), whereas the time-frequency patterns reflect *representational content*, a metric robust enough to survive removal of any particular subset of frequencies.

Reviewer #3:

*This study leverages intracranial EEG recordings from human participants to find evidence for reinstatement of trial-specific time-frequency patterns in the hippocampus during successful associative retrieval. Further, the degree of hippocampal reinstatement scales with gamma power increases and alpha power decreases. I appreciated the control analyses, including comparison to the average similarity of all successful retrieval events of the same target association ("AR surrogate"), computing similarity without gamma frequencies, and computing similarity collapsing across time, which strengthen the findings. Overall, the results are convincing evidence for hippocampal reinstatement in the time-frequency domain, though interpretation of reinstatement in the frequency (rather than spatial) domain is needed.*

*Major comments:*

*1) Defining a pattern across time/frequency vs. across space: Models of pattern completion generally assume that events are represented by sparse patterns of activity across neurons, and connections formed between these neurons during learning then facilitate activation of the full network of neurons given partial input. Based on this model, the pattern of activity that is "completed" during retrieval would be spatial in nature (i.e., across neurons). However, rather than utilizing the spatial data from channels along the long axis of the hippocampus, only one channel was selected, and the "pattern" of activity evoked by an event was in the temporal and frequency domain (and frequency information alone -collapsing across time - was sufficient to find the reported effects). Why is this the case?*

The reviewer raises a critical point here, and we should have been more explicit about our motivation to include only data from a single electrode. First, we need to clarify that even in the analysis that collapses the relative power changes across time prior to calculating reinstatement, the power values per se always contain temporal information given that spectral decomposition always requires data segments extended in time (for instance, the power value at 1 sec would differ between using a 200 ms vs. a 400 ms time window to extract the spectral signal). Thus, all our analyses inherently incorporate temporal dynamics of neural activity.

Nevertheless, we completely agree that pattern completion most likely requires a spatial component, and as pointed out by the reviewer, recent rodent findings confirm that reinstatement can be observed in local hippocampal neuronal circuits (Tayler et al., Current Biology 2013; Liu et al., Nature 2012). Critically, we would argue that the field potential picked up by an intracranial electrode reflects precisely the intricate local spatial (and temporal) constellation of underlying cell activity which is thought to show pattern completion. Indeed, in their recent review (Nature Reviews Neuroscience 2012), Buzsaki and colleagues make the compelling case that the amplitude and frequency of the extracellular field potential depend on the proportional contribution of diverse neuronal processes in the underlying volume (see also Agarwal et al., Science 2014). These considerations are now included in the Analysis section.

We would even go further to argue that including multiple hippocampal contacts spaced.5 cm apart would be too coarse an array to pick up local pattern completion processes. Of note, studies that relied on multi-electrode patterns to assess reinstatement (Jafarpour et al., 2014; Yaffe et al., 2014; Staudigl et al., 2015) did not record activity directly from the hippocampus, and it may well be the case that there exists a gradient in the spatial resolution of reinstatement between the hippocampus and cortical regions (but more work is clearly needed to test this idea).

Finally, to substantiate our claims, we conducted additional analyses including more electrode contacts. As elaborated in response to Point 2 below, we did not observe any notable increase in reinstatement when including posterior hippocampal contacts in each participant, consistent with the notion that – at least in our current paradigm – local (anterior) hippocampal pattern completion processes are well captured by single contact time-frequency profiles.

*The analyses investigating the effects of the gamma power increase on pattern reinstatement were a nice addition (and suggest that the similarity for AR trials is not driven solely by increased signal in the gamma band). However, while reinstatement was still significant within the time window of interest, would the increased similarity for AR > IR and AR > AR surrogate hold with cluster correction when excluding gamma frequencies? If not, it seems misleading to say that the "reinstated pattern per se might be largely carried by lower frequencies." Which frequency bands are contributing to the increased pattern similarity? It seems that the similarity information comes from the frequency bands (rather than a temporal pattern unfolding over time), given that the same pattern of results is found when averaging across time, and it might be informative to re-run analyses excluding certain bands to see whether a given band has a differential contribution to the pattern similarity.*

We thank the reviewer for raising this important point, which resonates with other reviewers’ question about the contribution of each frequency band. We now addressed this issue in a systematic and analytical fashion. First, we re-conducted the analyses (i) relying on all frequency bands but one, systematically excluding one band after the other, and (ii) relying on one frequency band only. We selected the following bands: delta (2-4 Hz), theta (4-8 Hz), alpha (8-12 Hz), beta (12-30 Hz) gamma 1 (30-50 Hz) and gamma 2 (50-100 Hz). We then compared both (i) the reinstatement values for AR trials and (ii) the conjoint effect of AR > AR surrogates and AR > IR to see whether a particular band is necessary (exclusion analysis) or sufficient (band only analysis) for the observed results. Second, to capture the contribution of each frequency (without binning frequencies into predefined bands), we analytically decomposed the correlation coefficient into the summed product of standard scores. This product would normally be summed across both the 41 time bins and the 43 frequency bins to yield the correlation coefficient, but by leaving the frequency component un-summed one can appreciate the relative contribution of each frequency to the final correlation (‘correlation weights’).

In both analyses, results suggest that no single frequency band would be necessary or sufficient to carry the reinstatement effect. And although null effects have to be interpreted with caution, this result aligns well with our intuition that event-specific reinstatement capitalizes on trial-by-trial variability in spectral contributions, drawing on much larger time-frequency information than could be contained in a single frequency/band. We now report this finding in the manuscript (Figure 2—figure supplement 1) and removed the statement that the reinstated pattern per se might be largely carried by lower frequencies. In addition, we now emphasize that the surrogate analysis actually already controls for potential signal-to-noise confounds, as AR surrogates have exactly the same spectral properties as the real AR bin.

*2) Selection of hippocampal channels: Why was only one electrode selected per participant? There were 7 subjects with a longitudinal implantation scheme, allowing for ~6-8 contracts along the long-axis of the hippocampus; given this data, it seems that the patterns could have been calculated across space, which would be more consistent with previous notions of "pattern completion." Secondly, why was the chosen electrode selected in the anterior hippocampus? Given some literature suggesting that posterior hippocampus is involved in retrieval, particularly of spatial information, and given that half of the associates were scenes, why weren't posterior hippocampus electrodes chosen (or at the very least, why were they excluded)?*

We pursued the strategy of selecting 1 contact per participant in order to conduct proper random effects analyses across our entire sample while being able to draw conclusions about the role of a particular brain region (in our case, the anterior hippocampus). Importantly, the preference for anterior vs. posterior hippocampal contacts stems, on the one hand, from the consistent observation of associative memory effects in that region in the neuroimaging literature (for review, see e.g. Davachi, 2006). Indeed, a recent opinion paper (Poppenk et al., 2013) specifically postulates a role of the anterior hippocampus in pattern completion (whereas the posterior hippocampus might be biased towards pattern separation). We now added this reference to the manuscript. On the other hand, a subsample of patients used in the current study showed strong associative recognition ERP effects in the anterior hippocampus (Staresina et al., 2012), which we now explicitly state in the Methods (Subsection “Electrode selection”).

That said, we followed the reviewer’s suggestion and conducted the reinstatement analysis in the 7 participants who had electrode coverage along the longitudinal axis of the hippocampus. There were on average 6.7 hippocampal contacts available (range: 5-8). Reinstatement was quantified as before, with power values averaged across the 400 ms time window and with channels as an additional feature dimension (as in Yaffe et al., 2014). The correlation was thus based on 43 x ~7 (frequency x channel) features.

In fact, the correlation values are numerically larger when using only the anterior hippocampal contact (0.5-1.5s encoding to 0.5-1.5s retrieval; t(6) = 2.32, P =.06), suggesting that inclusion of more widespread hippocampal contacts doesn’t benefit detection of pattern completion, but in fact might even add noise to the correlations. We note however that interpretive caution is warranted as this result is based on 7 participants only.

Author response image 3.The figure shows that the resulting reinstatement map for AR trials based on all hippocampal channels (left) is qualitatively very similar to the same participants’ reinstatement map for AR trials based on the selected anterior hippocampal contact only (right).**DOI:**
http://dx.doi.org/10.7554/eLife.17397.017

Lastly, we directly examined potential reinstatement effects in the posterior hippocampus, selecting the most posterior hippocampal contact in each participant. Applying the criteria reported in the manuscript, there was no evidence for pattern completion in the posterior hippocampus: neither the conjunction of AR > IR & AR > AR surrogates nor each of these comparisons individually survived cluster correction.

*3) Response time should be reported: It would be helpful to see the response time data. Are there differences in RT between color and scene trials? Is there a difference in RT between "don't know" and "association incorrect" trials?*

There were no differences between color and scene trials, nor between “don’t know” and “association incorrect” trials, further justifying collapsing across those trial types to increase statistical power. The RTs for “?” and incorrect target association responses are now reported in the main Results section:

“Response latencies were significantly shorter for AR compared to IR (1.91 s (+/-.13 s) vs. 2.14 s (+/-.12 s), t(10) = 2.88, P =.016). When considering “?” responses and incorrect target responses separately, their response latencies did not differ reliably (t(10) = 1.12, P =.290), whereas correct AR responses were significantly faster than both (both t(10) > 2.29, P <.05).”

The RTs for color vs. scene trials are now added to the main Results section in which the similarities of color and scene trials are reported:

“Finally, when including the factor Category (colors, scenes) and Memory (AR, IR) in a repeated measures ANOVA on response latencies, there was only a main effect of Memory (F(1,10) = 6.75, P =.027), without a Category main effect (F(1,10) = 1.54, P =.243) or a Category x Memory interaction (F(1,10) = 0.68, P =.428).”

*4) What is the degree of pattern reinstatement for IR trials? Given that approximately 1/3 of the "IR" trials are "association incorrect" trials (during which time participants may be reinstating another event, which could lead to increased dissimilarity relative to the "don't know" trials), it might also be informative to calculate pattern similarity across "don't know" and "association incorrect" trials separately. In addition, looking at the plot of IR reinstatement (Figure 2), it appears that there may be a cluster of significant reinstatement of 1-1.5s encoding at <1s retrieval; is this significant with an IR > IR surrogate contrast?*

The cluster of IR reinstatement of 1-1.5s encoding at <1s retrieval did not survive statistical comparison with IR surrogates, neither at the map level nor in direct comparison from 1-1.5s encoding to.5-1s retrieval (t(10) = 0.78, P =.451).

We agree with the reviewer that splitting up IR trials into IR_?_ and IR_incorrect_ would be an interesting analysis. However, we are afraid our paradigm is not ideally suited for such an analysis. There was a considerable range in how participants used the “don’t know” option, with such response variability being quite common to patient studies. Two patients would have less than 10 trials in the IR_incorrect_ bin and another 3 patients would have less than 10 trials in the IR_?_ bin, precluding us from doing robust statistical comparisons with those conditions.